# Molar Septum Expansion with Osseodensification for Immediate Implant Placement, Retrospective Multicenter Study with Up-to-5-Year Follow-Up, Introducing a New Molar Socket Classification

**DOI:** 10.3390/jfb12040066

**Published:** 2021-11-25

**Authors:** Samvel Bleyan, João Gaspar, Salah Huwais, Charles Schwimer, Ziv Mazor, José João Mendes, Rodrigo Neiva

**Affiliations:** 1Private Practice, 3/1 Tatarskaya St., 115184 Moscow, Russia; samvel32@gmail.com; 2Clinical Research Unit (CRU), Centro de Investigação Interdisciplinar Egas Moniz (CiiEM), Egas Moniz Cooperativa de Ensino Superior, 2829-511 Almada, Portugal; jmendes@egasmoniz.edu.pt; 3Department of Periodontics, School of Dental Medicine, University of Pennsylvania, Philadelphia, PA 19104, USA; shuwais@versah.com (S.H.); f40chuck@gmail.com (C.S.); rneiva@upenn.edu (R.N.); 4Department of Periodontics, School of Dental Medicine, University of Pittsburgh, Pittsburgh, PA 15260, USA; 5Private Practice, 144 Begin St., Tel Aviv 6492102, Israel; mazor2@yahoo.com; 6Department of Periodontics, Titu Maiorescu University, 040441 Bucharest, Romania

**Keywords:** osseodensification, immediate implant placement, septum expansion, osteotomy, osseointegration

## Abstract

The ideal positioning of immediate implants in molar extraction sockets often requires the osteotomy to be in the interradicular septum, which can be challenging in some cases, with traditional site preparation techniques. Patients who had undergone molar tooth extraction and immediate implant placement at five different centers, and followed up between August 2015 and September 2020, were evaluated. Inclusion criteria were use of the osseodensification technique for implant site preparation. The primary outcome was septum width measurement pre-instrumentation and osteotomy diameter post expansion. Clinical outcomes, such as implant insertion torque (ISQ) and implant survival rate, were also collected. A total of 131 patients, who received 145 immediate implants, were included. The mean overall septum width at baseline was 3.3 mm and the mean osteotomy diameter post instrumentation was 4.65 mm. A total of ten implants failed: seven within the healing period and three after loading; resulting in a cumulative implant survival rate of 93.1%. This retrospective study showed that osseodensification is a predictable method for immediate implant placement with interradicular septum expansion in molar extraction sockets. Furthermore, it allowed the introduction of a new molar socket classification. In the future, well-designed controlled clinical studies are needed to confirm these results and further explore the potential advantages of this technique.

## 1. Introduction

Immediate implant placement (IIP) into fresh extraction sockets has aroused interest since it was initially described [1] and has been considered a predictable therapeutic approach for both anterior and posterior sites, with survival rates comparable to implants placed in healed ridges [2,3,4,5]. An 11-year retrospective study of 300 implants immediately placed in molar extraction sockets reported an overall survival rate of 97.3% [6]. Furthermore, a systematic review [7] of outcomes following immediate molar implant placement demonstrated a survival rate of 98%, with no significant differences between maxilla and mandible. More recently, another systematic review and meta-analysis [8] of immediate implants in molar extraction sites demonstrated success rates of 93.3% after 1 year of follow-up.

This treatment alternative offers several advantages in comparison to the classic delayed approach, namely a single surgical intervention, with a reduction in overall treatment time and, therefore, increased patient satisfaction [9,10]. However, its success was reported to be influenced by several factors, including the need for atraumatic extraction to preserve favorable socket anatomy, as well as the effect of site instrumentation to achieve an adequate initial implant stability [9]. Implant primary stability and adequate insertion torque are considered critical aspects for successful IIP [5,6,11,12]. Several challenges have been described in achieving initial stabilization in molar extraction sockets. These include the width of the extraction socket, poor bone quality, inadequate interradicular bone septum width, and anatomical limitations beyond the apex of the roots, such as the inferior alveolar canal in the mandible or the maxillary sinus in the maxilla [13]. Thus, flapless tooth extraction with minimal trauma and gentle separation of the roots is essential to preserve a favorable anatomy and to allow the placement of the implant within the socket itself, when needed [10,14]. In addition, implant primary stability and insertion torque is related to the density of the bone pre and post site preparation. Bone density is known to have a direct effect on implant stability, as the denser the bone surrounding the osteotomy walls, the higher the insertion torque and the ISQ values [15]. Both these parameters are influenced by the drilling protocol [16,17], so enhancing the bone density during osteotomy preparation may improve clinical success, especially in the maxilla, due to its typically lower bone density compared to the mandible [18].

Smith and Tarnow [14] classified molar sockets based on the amount of interradicular septal bone in relation to implant placement into three types: Type A sockets have sufficient septal bone bulk to circumferentially contain the implant. Type B sockets have enough septal bone bulk to stabilize the implant, but not fully surround it. On the other hand, Type C sockets have insufficient septal bone to stabilize the implant without engaging the socket walls, so this would either indicate the placement of ultra-wide diameter implants or a delayed placement approach. According to several authors [19,20,21], immediate implant placement in molar extraction sockets using ultra-wide implants demonstrates a predictable outcome, with reduced bone loss and stable soft and hard tissue conditions. However, in a systematic review [7] conducted in 2016, ultra-wide implants (>6–9 mm) were found to have a significantly higher failure rate than implants of 4 to 6 mm diameter. More recently, Ragucci et al. [8] also recommended the use of implants of <5 mm diameter for immediate placement in molar extraction sockets. Therefore, implant placement in the interradicular septum is usually considered the best option for an immediate molar implant, not only in terms of correct 3D positioning, but also regarding implant survival [10].

Recently, a novel non-subtractive surgical technique for implant site preparation termed osseodensification (OD) was introduced [22]. Contrary to traditional extractive drilling protocols, it preserves bone and enhances its plasticity, utilizing specially designed burs that rotate in a non-cutting (counter-clockwise) direction to gradually expand the osteotomy, while simultaneously compacting bone into its trabecular spaces, increasing the density of the site [22,23,24,25]. Furthermore, OD was shown to enhance implant primary stability, due to the compaction auto-grafting and the associated spring-back effect [22,26]; increasing bone-to-implant contact (BIC) upon implant placement [24,25]. These autografted bone particles in the trabecular spaces act as nucleation for faster bone formation around the implant, potentially shortening the healing time [23,24,25,27]. Large-animal histological studies have demonstrated that this high stability at the day of surgery is maintained throughout the implant healing process, regardless of the implant macro- or micro-geometry [24,25,27]. In a recent multicenter controlled clinical trial, OD also demonstrated significantly higher insertion torque and ISQ values compared to conventional subtractive drilling for all implant dimensions, with the exception of short implants, regardless of the jaw and area operated, and irrespective of the evaluation period [28]. Osseodensification’s ability to plastically expand trabecular bone with compaction autografting, to facilitate implant placement with sufficient stability and adequate healing in sites with less than optimum bone quantity and quality, was documented in both in vivo and clinical data [23,29]. Trisi et al. [23] was able to demonstrate, in a large animal histological study, the predictability of placing a 5-mm implant in 5-mm wide ridge in sheep iliac crest with adequate healing. Koutouzis and Huwais [29] confirmed his findings in a clinical controlled study that demonstrated a 93% success rate for 38 implants placed in plastically expanded alveolar ridges via osseodensification in 21 patients. In addition to ridge plastic expansion, osseodensification has also been reported to enhance dental implant’s short and long-term success rate, regardless of their macro- or micro-geometry, in several clinical scenarios, including immediate loading [30,31,32], as well as to facilitate implant placement in conjunction with crestal sinus graft, with a high success rate [33,34,35].

The aim of the present multicenter retrospective study was to assess the effectiveness of interradicular septum expansion with osseodensification site preparation for immediate implant placement in molar extraction sockets.

## 2. Materials and Methods

This retrospective analysis followed the World Medical Association Declaration of Helsinki and the directives given by the Egas Moniz Ethics Commission (CEEM) at Egas Moniz Cooperativa de Ensino Superior, Monte de Caparica, Portugal, which does not require ethical approval for retrospective clinical studies.

An informed consent form was signed by all patients included in the study, both for the clinical procedure and follow-up appointments. All treatment steps and data collection were part of the routine procedures at the centers, and no extra measures were taken for the purpose of the study. All examiners were blind, since a random case number was allocated to the extracted data, ensuring patient anonymity and data protection. The study was structured following the STROBE statement [36].

### 2.1. Selection Criteria and Surgical Technique

Patients who had undergone molar tooth extraction and immediate implant placement with osseodensification at five different centers (S.B., J.G., S.H., C.S., Z.M.), followed up between August 2015 and September 2020, were evaluated. Inclusion criteria included patients with molar extraction sockets that had an interradicular septum of at least 2.5 mm width, use of the osseodensification technique for implant site preparation, and follow-up of a minimum of 12 months after loading with a definitive implant-supported restoration. Exclusion criteria comprised an initial septum width <2.5 mm, history of radiotherapy, bisphosphonate medication, active periodontal disease, uncontrolled diabetes, heavy smoking (>20 cigarettes/day), and local acute apical abscess. All patients had a cone beam computed tomography (CBCT) prior to surgical procedure.

All interventions were performed by experienced surgeons, who followed standardized surgical technique. After local anesthesia with articaine (4%) and epinephrine (1:200,000), flapless tooth extraction, as atraumatic as possible, was performed after separation of the roots with a long thin diamond bur, in order to preserve the interradicular bone and the general socket anatomy. The socket was then thoroughly curetted to detach any granulation tissue that could potentially impair healing.

Septum width was directly measured post molar extraction. Measurement was recorded at the narrowest width of the septum. Implant site preparation started with a pilot drill, in clockwise motion, in the center of the septum, until 1 mm deeper than the planned implant length. Densah^®^ Burs (Versah, LLC, Jackson, MI, USA) were then sequentially used in OD mode (counterclockwise, drilling speed 800–1500 rpm, with copious irrigation) in small increments to gradually expand the osteotomy, until reaching the desired width for the planned implant diameter (Figure 1).

Osteotomy diameter as a reflection of septum width expansion was then directly measured and recorded after site instrumentation. Although each center used the implant company of their choice, all implants placed were conical, bone-level, and with internal connection (Table 1). After implant placement at the adequate depth, the gaps were filled with allograft or alloplastic, depending on each center’s preference and either a customized or a large stock sealing healing abutment was placed, with no attempt to coronally advance the flaps for primary intention healing. The insertion torque value was registered, and implant stability was measured using resonance frequency analysis, immediately after implant insertion (primary stability) and after healing, before final impression (secondary stability).

The osseointegration period varied according to the decision of each clinician, based on the records mentioned above and on bone quantity and quality, with a minimum of 3 months. Despite not following a standardized medication protocol, all patients were prescribed post-operative antibiotics for 7–10 days, based on each center’s preference.

### 2.2. Variables and Statistics

Data regarding patient characteristics (age and gender); tooth location; date of surgical and restorative procedure; septum width before and after site preparation and expansion; insertion torque and ISQ at baseline and after osseointegration; implant width and length; time of loading; osseointegration success rate; and final follow-up appointment were collected from the patients’ clinical files. The primary outcome was septum width measurement pre-instrumentation and osteotomy diameter post expansion. Descriptive statistics were conducted using IBM^®^ SPSS^®^ Statistics software (SPSS for Mac, Version 26.0. SPSS Inc. Chicago, IL, USA). A Kaplan–Meier curve was used to analyze the survival rate of implants placed. This curve was adjusted to 12 months, because it was the minimum follow-up common to all implants.

## 3. Results

A total of 131 patients, 90 women and 41 men, with a mean age of 52 years (range 27–80), who received 145 immediate implants in molar extraction sockets, were included (Figure 2). The mean follow-up of the included patients was 36 months (range 12–60 months). Reasons for tooth extraction were endodontic treatment failure, root fracture, or non-restorable teeth. No extracted teeth sockets for periodontal reasons were included.

A total of 87 implants were placed in the mandible (72 in first molar sites and 15 in second molar sites) and 58 in the maxilla (53 in first molar sites and 5 in second molar sites), as shown in Figure 3. Maxillary sockets had higher mean values of interradicular septum width compared to those in the mandibular, as described in Figure 4.

The mean overall septum width at baseline was 3.3 mm, and the mean osteotomy diameter post instrumentation was 4.65 mm after expansion with osseodensification (Figure 5).

Implant stability was measured by both insertion torque (ITV) and ISQ values. ITV was higher in the mandible (mean 46.72 N cm; range 30–60 N cm) than in the maxilla (mean 41.12 N cm; range 20–60 N cm), with an overall mean value of 44.48 ± 8.2 N cm (Figure 6).

Only 6.2% of the implants had an ITV <3 5 N cm, while 35.9% had an ITV ≥ 50 N cm. Mean ISQ was 72.8 (range 60–82) at baseline on the day of surgery (ISQS) and 78.9 (range 70–88) after the osseointegration period, before final impression (ISQR), as described in Table 2. Implant diameter ranged from 4.2 to 6.4 mm, and length ranged from 10 to 13 mm, depending on the implant system used in each center. A total of ten implants (four in the mandible and six in the maxilla) failed (Table 3): seven within the healing period before final impression and three after loading, resulting in a survival rate of 93.1%. Only two centers included smoker patients (*n* = 6), who did not experience implant failure; therefore, no correlation could be assessed between smoking and implant failure. The Kaplan–Meier estimator predicted a 93.1% survival rate at 12 months follow-up (Figure 7).

## 4. Discussion

According to a recent systematic review and meta-analysis [8], the suggested approach for IIP in molar extraction sockets includes a flapless procedure, a one-stage implant placement, grafting the gap, and the use of implants with <5 mm diameter. Flapless surgery may not only contribute to decreased operative time, but also to faster healing, reduction of peri-implant tissue collapse, less postoperative complications, and improved patient comfort [8]. This approach was followed by all centers in this study, except for the implant diameter. In fact, 64.2% of the implants placed by all centers had a diameter ≥ 5 mm. The mean overall septum width at baseline was 3.3 mm and the mean implant diameter of all implants placed was 4.96 mm, which demonstrates the potential of the osseodensification technique to preserve the bony housing and expand the septum; thus, allowing predictably placing wider diameter implants compared to the conventional osteotomy technique.

Walker et al. [11] assessed the relationship between insertion torque values and clinical outcomes and reported an implant survival rate in immediate molar implants of 86%, when insertion torque was low, and 90% to 96% when IT was medium to high, respectively. This tendency was also observed in our study, since four out of the nine implants that had an insertion torque < 35 N cm failed. Moreover, the mean insertion torque of the implants that failed (*n* = 10) was 37.5, while the mean insertion torque of the successful implants (*n* = 135) was 45. Regarding ISQ, implants that ended up failing had lower mean ISQ values (68.3) on the day of surgery, compared to implants successfully integrated and loaded (73.2). In a recently published multicenter controlled clinical trial [28], OD drilling demonstrated significantly higher IT and temporal ISQ values relative to more conventional subtractive drilling techniques for all implant dimensions, with the exception of short implants. Therefore, we may assume that the implant survival would probably be lower with a traditional drilling protocol.

All sockets evaluated in this retrospective analysis were grafted with either allograft or alloplast (Novabone^®^). Bone grafting of the remaining socket voids adjacent to an immediate implant is not essential for osseointegration to occur, especially if the outer walls of the socket are intact [10,37,38]. However, its combination with a customized healing abutment, acting as a prosthetic socket seal device minimizes the amount of ridge contour change after tooth extraction and IIP, thereby contributing to better esthetics and restorative contour [10,38], as observed in this study (Figure 8).

Pre-operative CBCT is an essential and effective diagnostic method to evaluate socket anatomy and to define the most suitable treatment approach for each case, as well as minimizing the risk of damaging vital structures [39]. Historically, a minimum 3 mm width of interradicular septal bone (ISB) was deemed important to achieve initial stabilization of an immediate molar implant [40]. In this study, twenty-three extraction sockets had an ISB width of 2.5 mm and one had 2.8 mm. Nevertheless, the osseodensification technique used for implant site preparation allowed adequate septum expansion of all these sites, to create osteotomies diameters in a range of 3.5–4.5 mm, thereby providing adequate implant stability upon insertion. Moreover, since it pushes bone in a both lateral and apical direction, instead of removing it, osseodensification also predictably allows sinus elevation using a crestal approach in maxillary molar sockets with reduced residual bone height below the sinus floor [33].

Traditionally, Smith and Tarnow classification type B sockets [14] with narrow septa are commonly managed by clinicians through the placement of an immediate implant into the palatal root socket of maxillary molars or into one of the two mandibular molar root sockets [41]. However, this may lead to potential food impaction and tissue inflammation, due to poor emergence profile of the restoration. Furthermore, Smith et al. [42] observed, in a retrospective study of 300 implants, that there is a direct correlation between the horizontal implant–tooth distance and the incidence of adjacent tooth decay. Therefore, immediate implant placement in the mesial or distal molar root sockets may significantly increase the risk of decay in the furthest tooth. Accordingly, the ideal implant positioning in molar sockets will most often require the osteotomy to be in the septum. Osseodensification may facilitate the preservation and the expansion of the interradicular septum, thus enhancing the ability to predictably place implants with adequate stability in both type B and C sockets, as shown in this study (Figure 9).

The traditional classification of molar extraction sockets by Smith and Tarnow [14] is based on the amount of interradicular septal bone remaining post instrumentation around immediately placed implants, but it does not take into account the specific measurement of the septum width pre-instrumentation nor pre-implant placement. Furthermore, the specific type of the socket in this classification is dependent on the diameter of the implant placed. Therefore, the authors of the present study propose a new diagnostic classification that is based on the initial septal bone width prior to site preparation and implant placement, which would allow adequate treatment planning. The new classification (Figure 10) includes four categories: S-I: septum initial width >4 mm; S-II: septum initial width = 3–4 mm; S-III: septum initial width = 2–3 mm; and S-IV: septum initial width < 2 mm/no septal bone. The relevance of this new diagnostic classification is related to the fact that, with osseodensification instrumentation, and due to bone preservation and plastic expansion, it is possible to convert type B sockets into type A, and type C into type B [14], as was observed in this study. According to our classification, only S-IV sockets represent an exclusion criterium for septum expansion with osseodensification. This would either indicate the placement of ultra-wide implants or a delayed placement approach. In fact, our results showed that osseodensification allowed immediate implant placement in the first three categories (S-I, S-II, and S-III) of this new classification, with adequate implant stability.

## 5. Conclusions

This up-to-5-year follow-up retrospective study showed that osseodensification is a viable and predictable method for interradicular septum expansion and immediate implant placement with adequate stability in molar extraction sockets. Furthermore, it allowed the introduction of a new molar socket classification, based on the available septum width prior to instrumentation. In the future, well-designed controlled clinical studies are needed to confirm these results and further explore the potential advantages of this site preparation technique.

## Figures and Tables

**Figure 1 jfb-12-00066-f001:**
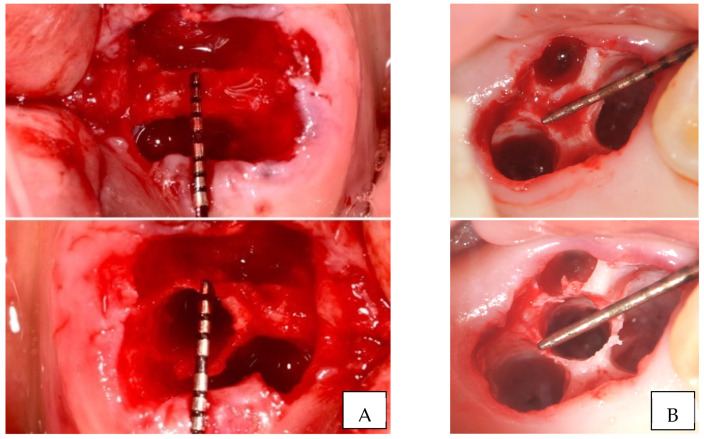
Clinical examples of interradicular septum expansion after implant site preparation with osseodensification ((**A**). Mandibular first molar; (**B**). Maxillary second molar).

**Figure 2 jfb-12-00066-f002:**
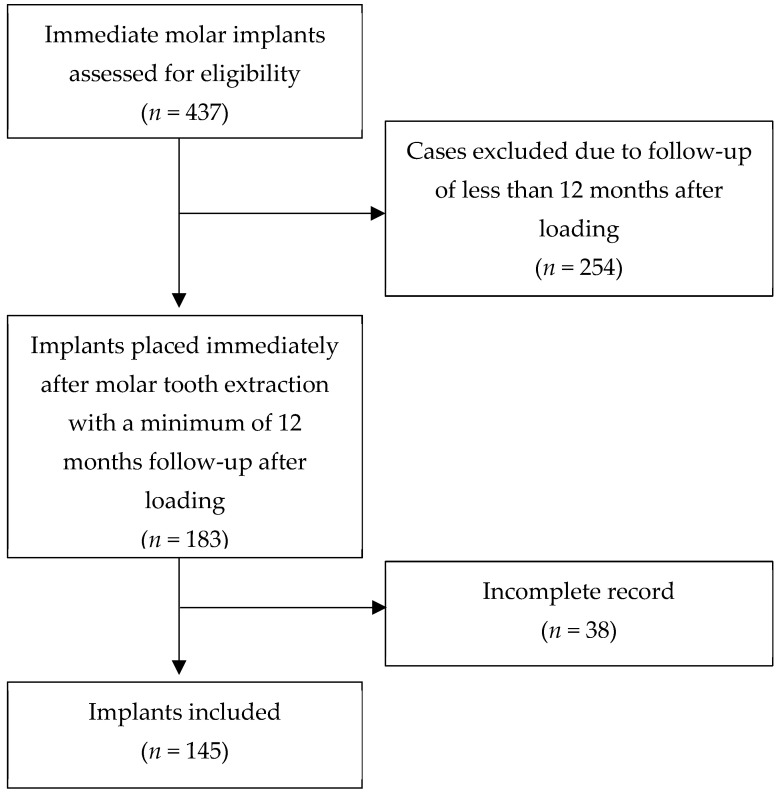
Flowchart representative of implants included in the retrospective analysis.

**Figure 3 jfb-12-00066-f003:**
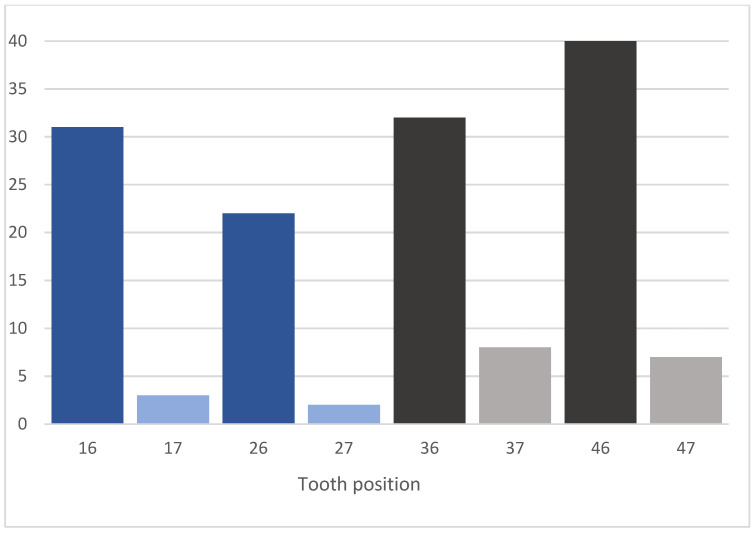
Number of implants placed, according to tooth position.

**Figure 4 jfb-12-00066-f004:**
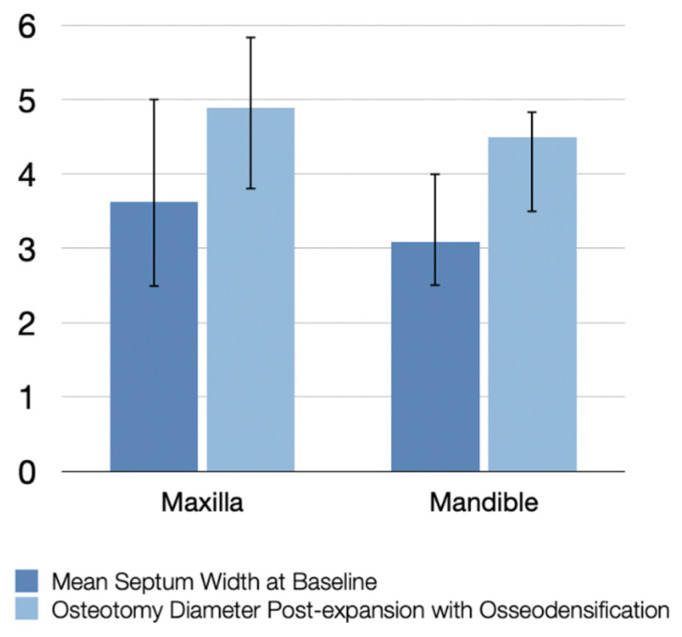
Mean septum width before instrumentation and osteotomy diameter post-expansion with osseodensification.

**Figure 5 jfb-12-00066-f005:**
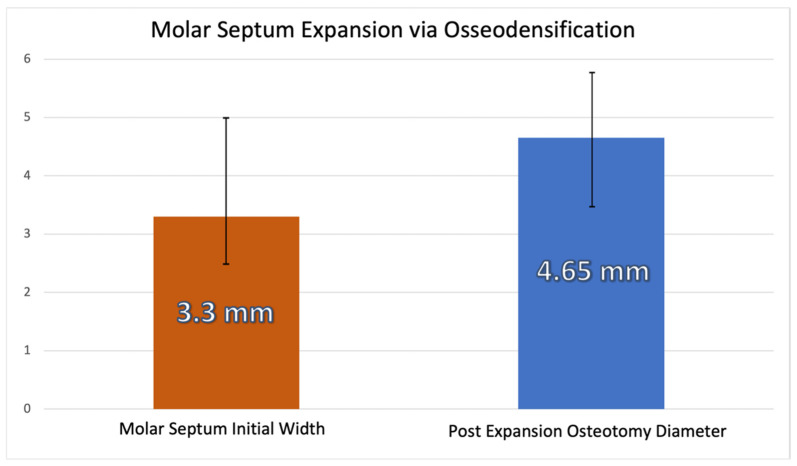
Mean overall septum width at baseline and osteotomy diameter post-expansion with osseodensification.

**Figure 6 jfb-12-00066-f006:**
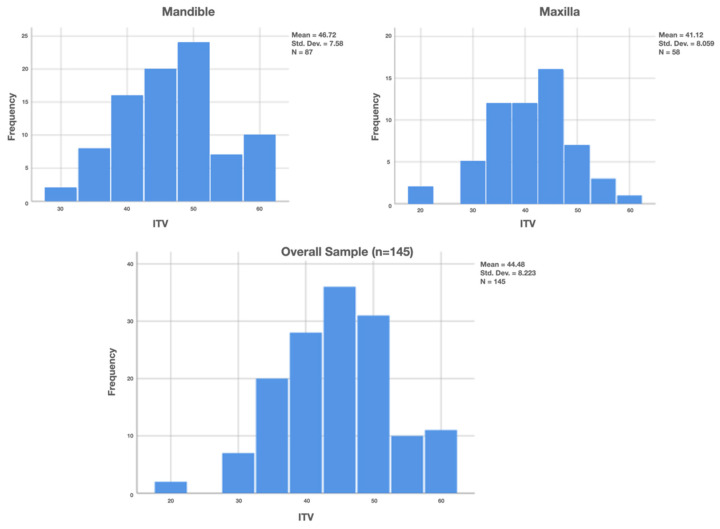
Insertion torque value (ITV) of implants placed.

**Figure 7 jfb-12-00066-f007:**
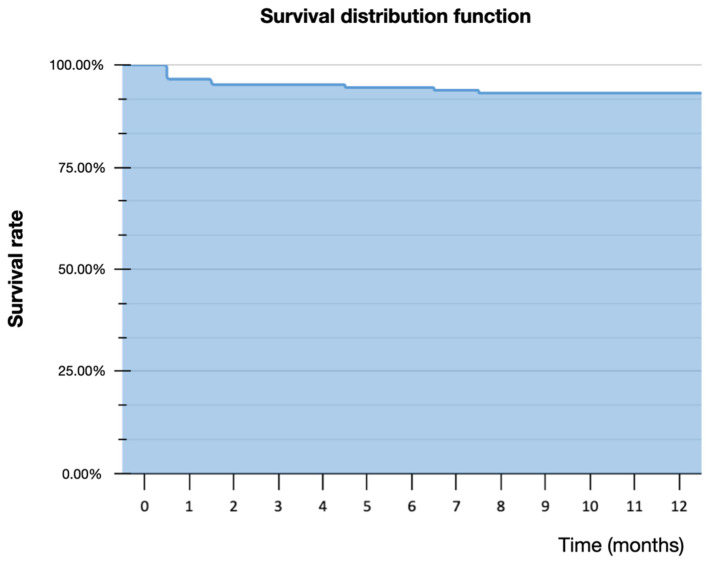
Kaplan–Meier survival curve for survival estimate.

**Figure 8 jfb-12-00066-f008:**
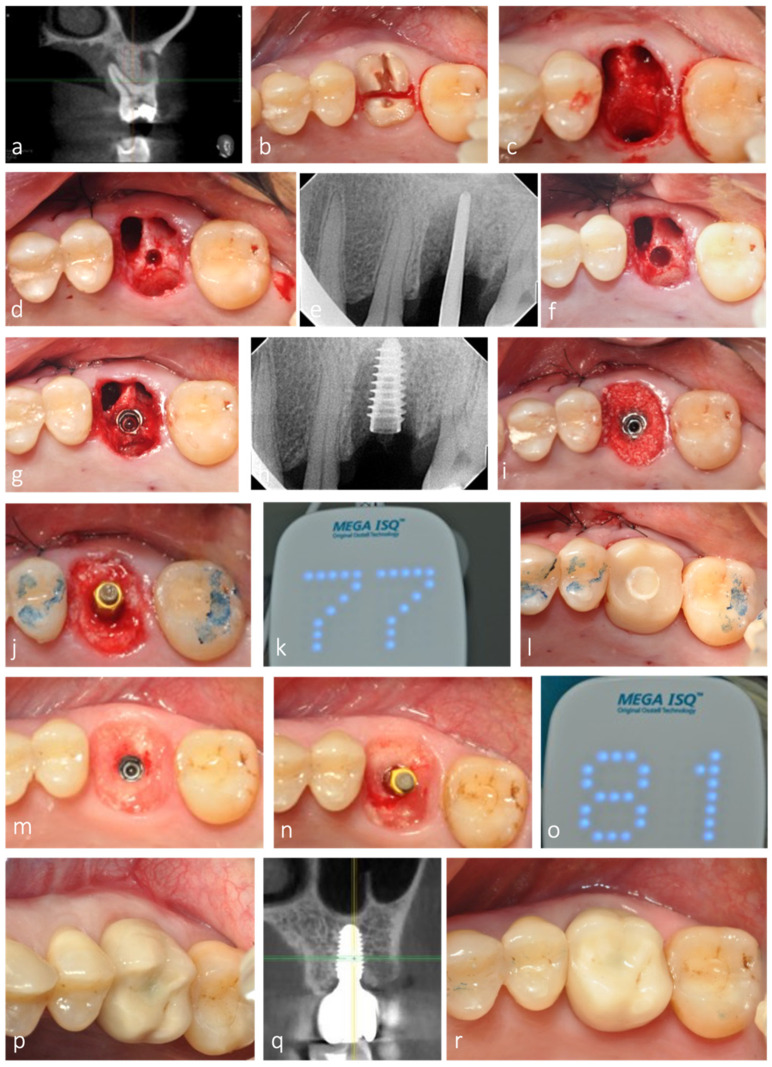
Representative clinical case with 3-year clinical and radiographic follow-up. (**a**) CBCT of maxillary left first molar showing periapical infection with extensive bone loss buccally and palatally. (**b**) Root section for tooth extraction as atraumatic as possible. (**c**) Septum preservation after extraction. (**d**,**e**) Initial osteotomy depth at 10 mm. (**f**) Implant site preparation, optimized with osseodensification. (**g**,**h**) Implant placed in the expanded septum. (**i**) Allograft placed in the root sockets to fill the extraction socket. (**j**–**l**) Adequate implant stability allowed for the placement of a fully contoured customized socket sealing healing abutment out of occlusion. (**m**) Healing after 3 months, with contour maintenance. (**n**,**o**) ISQ measurement after osseointegration period. (**p**–**r**) Clinical and radiographic follow-up after 3 years.

**Figure 9 jfb-12-00066-f009:**
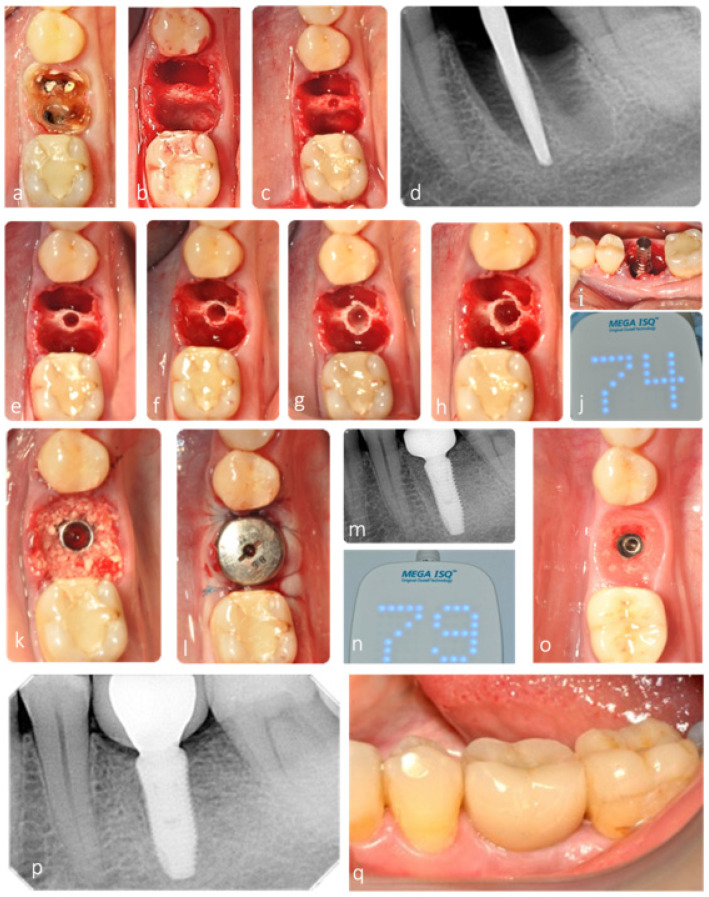
Representative clinical case with 4-year clinical and radiographic follow-up. (**a**) Initial radiograph with 4 mm of residual alveolar bone height. (**a**) Clinical situation at baseline. (**b**) Occlusal view after gentle tooth extraction with maintenance of interradicular septum. (**c**–**h**) Septum expansion after sequential instrumentation with osseodensification. (**d**) Radiograph of densifying bur VT1525 (2.0) in interradicular septum. (**i**,**j**) ISQ measurement after implant placement (primary stability). (**k**–**m**) Grafting of the gap and socket sealing with large healing abutment. (**n**) ISQ measurement after osseointegration period (secondary stability). (**o**) Contour maintenance after healing. (**p**,**q**) Clinical and radiographic follow-up after 4 years.

**Figure 10 jfb-12-00066-f010:**
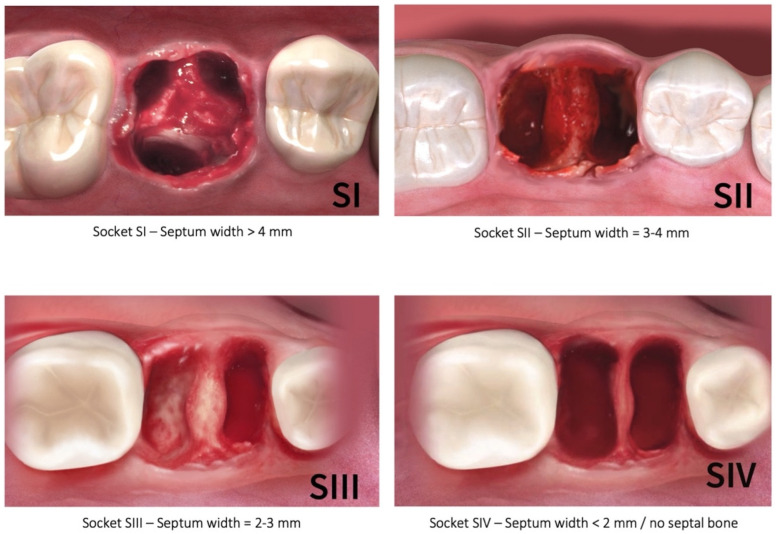
New molar socket classification according to the initial interradicular septum width. (SI—septum width > 4 mm; SII—septum width = 3–4 mm; SIII—septum width = 2–3 mm; SIV—septum width < 2 mm/no septal bone).

**Table 1 jfb-12-00066-t001:** Overview of implants included in the retrospective analysis.

Implant Company	Number of Implants Placed	Number of Implants Failed
Dentium	35	1
Adin	35	3
Megagen	26	1
Neobiotech	21	2
Zimmer	14	3
Paltop	6	0
IDI	5	0
Nobel Biocare	3	0
Total	145	10

**Table 2 jfb-12-00066-t002:** Mean ISQ measurement on day of surgery and in restorative phase after osseointegration period.

		ISQS	ISQR
Maxilla	Mean	71.47	77.26
	N	58	54
	Std. Deviation	4.231	3.004
Mandible	Mean	73.72	79.88
	N	87	84
	Std. Deviation	4.358	3.730
Total	Mean	72.82	78.86
	N	145	138
	Std. Deviation	4.434	3.684

ISQS–ISQ in day of surgery; ISQR–ISQ in restorative phase.

**Table 3 jfb-12-00066-t003:** Description of failed implants.

Implant Company	Diameter	ITV	ISQ	Septum Pre	Septum Post	Timing of Failure
Neobiotech	5	55	76	3.5	4.8	After
Neobiotech	5	35	65	3.4	4.8	Before
Dentium	5	40	68	2.8	4.5	Before
Zimmer	5.2	20	63	5	5.5	Before
Zimmer	5	20	62	4	5.5	Before
Zimmer	4.7	30	60	2.5	4.5	After
Megagen	5.0	30	70	3.5	4.8	After
Adin	4.3	50	75	3	4.5	Before
Adin	5	45	70	3	4.5	Before
Adin	4.3	50	74	2.5	3.8	Before
Mean	4.85	37.5	68.3	3.32	4.7	

## Data Availability

Data can be requested to the corresponding author upon reasonable request.

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
