# Peer review of "Molar Septum Expansion with Osseodensification for Immediate Implant Placement, Retrospective Multicenter Study with Up-to-5-Year Follow-Up, Introducing a New Molar Socket Classification"

_jfb, 2021, doi:10.3390/jfb12040066_

Round 1

Reviewer 1 Report

I recommend that the change of the title of your study because the main subject of your manuscript was molar septum expansion utilizing osseodensification not to multicenter study up to 5 years f/u.

Even though it is retrospective study, approval of IRB should be performed. Any studies that used the clinical information is required the approval of your institute.

Figure 2,3, and 5 look blurry, change it more clear picture.

Is there any advantage this technique of osseodensification compared with conventional extractive drilling? If your study can be worthful and meaningful, the insertion torque and ISQ when using OD may be higher than conventional technique. So in discussion, you should describe the value of parameter of torque and ISQ when using conventional technique and compare with your study.

Reviewer 2 Report

The study is certainly interesting for the dentist who deals with implantology, but I believe that some variables could be investigated to make the study even more interesting if this is possible by the authors.
1. Format the tables according to the MDPI guidelines
2. it would be useful to represent the selection process of the patients included in the study by means of a flow chart
3. what type of variables were measured in the population included in the study? (age, sex, smoking, systemic diseases, etc.) and possibly represent them by means of a table.
4. Could these variables have influenced the implant failure and survival rate?
5. Could implant survival data be represented by a kaplan meier curve? The Hazzard Ratio could be calculated between the failures of implants placed in a site where there was the loss of a tooth due to endodontic failure and other causes. The question I ask myself is whether the endodontic failure that is often accompanied by an inflammatory apical lesion can influence implant success with the method you described?

Round 2

Reviewer 1 Report

Thanks for your revision and re submission.

I still concern about the ethical problem on your manuscript.

You said that in your institute judged your study was except in the target of evaluation

and you already have been cleared out from JFB Editorial Board and Editorial Assistance.

But as I have been known, the all of study that was used the patient clinical information such as clinical photo and

radiograph, and medical record was target of evaluation by IRB  

Author Response

We appreciate very much your important recommendations which were followed accordingly. We take this opportunity, as well, to thank you for the time and effort spent in this review process.

Reviewer 2 Report

The authors have made all the required changes, I believe the manuscript can be published in this forma...

minor review

1)the tables are not yet formatted according to the MDPI guidelines

Author Response

Dear Reviewer 2,

We appreciate your time and effort spent in this review process.

I believe that now the tables are according to the MDPI guidelines.

Yours sincerely,

João Gaspar

Clinical Research Unit (CRU) Department

Implantology Department

Egas Moniz - Cooperativa de Ensino Superior Caparica, Portugal

+351911909629

jgaspar@egasmoniz.edu.pt

j-gaspar@hotmail.com